# Trends in Bioactive Multilayer Films: Perspectives in the Use of Polysaccharides, Proteins, and Carbohydrates with Natural Additives for Application in Food Packaging

**DOI:** 10.3390/foods12081692

**Published:** 2023-04-19

**Authors:** Luisa Bataglin Avila, Carlos Schnorr, Luis F. O. Silva, Marcilio Machado Morais, Caroline Costa Moraes, Gabriela Silveira da Rosa, Guilherme L. Dotto, Éder C. Lima, Mu. Naushad

**Affiliations:** 1Research Group on Adsorptive and Catalytic Process Engineering (ENGEPAC), Federal University of Santa Maria, Av. Roraima, 1000-7, Santa Maria 97105-900, Rio Grande do Sul, Brazil; 2Department of Civil and Environmental, Universidad De La Costa, Calle 58 # 55-66, Barranquilla 080002, Atlantico, Colombia; 3Chemical Engineering, Federal University of Pampa, 1650 Maria Anunciação Gomes Godoy Avenue, Bage 96413-172, Rio Grande do Sul, Brazil; 4Graduate Program in Science and Engineering of Materials, Federal University of Pampa, 1650, Maria Anunciação Gomes de Godoy Avenue, Bage 96413-172, Rio Grande do Sul, Brazil; 5Institute of Chemistry, Federal University of Rio Grande do Sul, Porto Alegre 90010-150, Rio Grande do Sul, Brazil; 6Department of Chemistry, College of Science, King Saud University, Riyadh 12372, Saudi Arabia

**Keywords:** multilayer packaging, layer-by-layer, biopolymers, bioactive compounds, electrospinning

## Abstract

The harmful effects on the environment caused by the indiscriminate use of synthetic plastics and the inadequate management of post-consumer waste have given rise to efforts to redirect this consumption to bio-based economic models. In this sense, using biopolymers to produce materials is a reality for food packaging companies searching for technologies that allow these materials to compete with those from synthetic sources. This review paper focused on the recent trends in multilayer films with the perspective of using biopolymers and natural additives for application in food packaging. Firstly, the recent developments in the area were presented concisely. Then, the main biopolymers used (gelatin, chitosan, zein, polylactic acid) and main methods for multilayer film preparation were discussed, including the layer-by-layer, casting, compression, extrusion, and electrospinning methods. Furthermore, we highlighted the bioactive compounds and how they are inserted in the multilayer systems to form active biopolymeric food packaging. Furthermore, the advantages and drawbacks of multilayer packaging development are also discussed. Finally, the main trends and challenges in using multilayer systems are presented. Therefore, this review aims to bring updated information in an innovative approach to current research on food packaging materials, focusing on sustainable resources such as biopolymers and natural additives. In addition, it proposes viable production routes for improving the market competitiveness of biopolymer materials against synthetic materials.

## 1. Introduction

The definition of food packaging consists of a material capable of guaranteeing the quality, integrity, and safety of a product. In other words, it is a system that offers conditions for transporting, distributing, and storing the product from its production, in the industry, to the final consumer. Over the years, the use of synthetic plastics for food packaging has represented a notable environmental concern. This fact is related to the pollution they cause and also to the scarcity of fossil resources. Moreover, the current pandemic has raised such problems, as there has been a considerable increase in food deliveries and, consequently, in the consumption of packaged foods [1,2]. 

The perception of the harmful effects on the environment caused by the indiscriminate use of these materials and the inadequate management of the disposal of post-consumer waste have boosted efforts to redirect this consumption to bio-based economic models [1,3]. In this sense, using biopolymers to produce materials is a reality for packaging companies searching for technologies that allow these materials to compete with those from synthetic sources [4]. Moreover, in some countries, especially the most economically developed, there is a greater awareness of eco-friendly products. The main bioplastics-producing continents are Asia, North America, and Europe.

Various biopolymers can be used to develop a packaging material, such as proteins, lipids, and polysaccharides [5]. However, materials produced with biopolymers still have limitations regarding their use as food packaging due to their weak mechanical properties and presenting a poor barrier to oxygen and water when compared to conventional packaging [6,7,8,9]. An alternative to improve these characteristics is the development of multilayer films. In this type of system, each layer plays a specific role. Generally, multilayer active food packaging consists of three layers: an outer layer, a middle layer, and an inner layer. The first has strong barrier characteristics, thus preventing the bioactive compounds in the intermediate layer from being lost to the environment. The second contains the bioactive compound and must have good diffusion properties. Finally, the third layer has a controlled release profile of the active compounds for the packed food [10]. With this method, it is possible to obtain a film that can be produced by combining different biopolymers to include in a single structure the resulting properties of each biopolymer used [11]. Another important point of this technology is the possibility of trapping the bioactive compound in one of the layers of the film, conferring antioxidant and antimicrobial properties, for example, attracting more attention and interest from the food packaging industry [9,11].

Commercially, synthetic additives are more common than natural ones. However, although food agencies regulate these additives, they are often associated with harmful effects on human health. Carocho et al. [12] discussed the possible impact of different food additives on human health in their work. Calcium propionate stands out as an inducer of sleep disorders in children, nitrites as carcinogens, parabens as reproductive reducers in men, and the consumption of large amounts of foods treated with sulfites can be a cause of dermatitis and diarrhea, for example. Therefore, researchers have explored natural alternatives worldwide [13]. The literature reports various bioactive compounds’ natural resources, including agro-industrial by-products. Residues, rich in bioactive compounds, can be leaves, peels, pulp, and seeds. Thus, these compounds can have several applications, such as natural additives in food packaging. However, despite promising applications, industrial uses still face challenges related to the low stability of active compounds [14]. Therefore, many authors have been developing multilayer active food packaging to minimize this problem and enable their application on an industrial scale.

In this context, this review aims to bring updated information in the innovative form on current research of multilayer films as a promising route for the use of biopolymers in the production of food packaging material that is commercially competitive with that from petrochemical sources. This work also includes the variety of biopolymers and natural resources of bioactive compounds and strategies for producing biopolymer materials. In addition, the applications of these materials containing natural additives as active food packaging and trends in this sector were highlighted.

## 2. Recent Multilayer Packaging Developments

Multilayer films are an emerging technology in the food packaging sector. This trend is related to the ability of this type of material to improve performance using a single packaging system [15]. Different methods and biopolymers are used to produce this type of packaging while considering the basic functions of the packaging as protection against contamination, moisture, oxidation process, and microbial action [15,16].

Many biopolymers have been used to develop multilayer packaging and are reported in the literature. Normally, a combination of biopolymers is used to achieve the desired properties. An example of this can be observed in the study developed by Andrade Del Olmo et al. [17], which used polylactic acid and chitosan to develop a multilayer film for food packaging. Another example is proposed by [11] who produced active bilayer films using cassava starch and polyvinyl acetate (PVA) as biopolymers. In addition, other biopolymers are often used to develop this type of material, such as gelatin, potato starch, zein, and soy protein isolate [15,18,19,20].

In addition to the possibility of using different polymers to improve the mechanical and barrier properties, the multilayer system also allows the controlled release of active compounds, which may be added to the polymer matrix, thus favoring the performance of active packaging [21]. For this reason, researchers have been developing materials with different functions, making it possible to obtain promising products for use as active food packaging. Table 1 summarizes some works carried out on this topic.

Table 1 highlights some studies published about multilayer food packaging. With this, it is possible to observe the variety of bioactive compound sources that can be introduced to the polymeric matrix and the different properties that can be conferred on the final product. In addition, the important function of extra layers in the packaging system is also worth mentioning. In line with this, there was an increasing number of publications on multilayer food packaging and active multilayer food packaging in the period 2011 to 2021, according to the scientific database Web of Science. Therefore, it is possible to observe the importance and relevance of the study and development of multilayer packaging systems. However, until now, Brazil has appeared in this scenario only in research on multilayer food packaging, appearing in seventh place in the number of publications. This fact clears the importance of researchers focusing their efforts on studying and developing this material.

In this sense, Andrade et al. [21] compared the monolayer film properties with the multilayer film properties and concluded that adding PVA layers improved elasticity and barrier capacity. Xia et al. [20] studied the best formulation for developing multilayer films based on zein and gelatin incorporated with tea polyphenols. Mechanical and barrier analyses were performed on monolayer zein and gelatin films and multilayer films of zein/gelatin obtained at different formulations. With this study, the authors identified the advantages and disadvantages of individual biopolymer films and optimized them in a single-packed structure.

Cai et al. [23] compared mono and multilayer films and verified improvements in mechanical and barrier properties and the controlled release of bioactive compounds when the multilayer films were used. The authors mentioned that while the monolayer film showed a release during 72 h, the multilayer film showed a release during 312 h. This fact was attributed to the hydrophobic outer layer, which created a physical barrier. Therefore, the inner layer was not in direct contact with the release medium. With this, the importance of choosing the polymers composing the multilayer system becomes evident, uniting the different properties to obtain the optimized condition.

In the same line, Quiles-Carrillo et al. [25] compared the controlled release of gallic acid in the bilayer and multilayer polylactide films. They found that bilayer films can release gallic acid in the first few days, while multilayer films can do so over long periods. On the other hand, Ordoñez et al. [26] evaluated the antimicrobial properties of PLA monolayer, starch monolayer, and PLA-starch multilayer films incorporated with ferulic and cinnamic acids. The authors observed that the active starch monolayer film exhibits an antimicrobial property. However, the same was not observed in the monolayer PLA film, and the combination of the active layer of starch in the PLA structure also did not show effective microbiological inhibition. This fact was attributed to the low molecular mobility through the PLA layer. Therefore, the importance of choosing the biopolymers used in constructing the layers is evident.

## 3. Methods of Preparing Multilayer Packaging

Over the years, more and more technologies have been used in food packaging materials to extend the shelf life and safety of packed foods. An important strategy is to use multilayer films that show better preservation than monolayer films [27]. Although it can be obtained by combining different biopolymers, the adhesion between the layers is crucial in developing this material. This information is especially related to the fact that poor adhesion can cause a reduction in important properties such as barrier properties [28].

Moreover, food packaging has primary functions, such as achieving preservation and the safe delivery of food products from their production until their distribution, commercialization, and consumption. These functions are directly affected by the materials used and the production conditions of packaging materials since biopolymers are known to form materials with poor barriers and mechanical properties compared to synthetic polymers [24,29]. At this point, it is important to know the characteristics of biopolymers and the properties of the materials developed to meet the requirements of food packaging. In general, for food packaging applications, mechanical properties such as tensile strength and elongation at break, barrier property (water vapor permeability and gas permeability), water solubility, optical properties, and thermal stability are evaluated [30,31].

In this sense, the multilayer system allows the combination of different biopolymers to optimize the properties described above. Thus, a range of possible packaging systems is described in the literature, each type requiring a specific preparation method.

### 3.1. Layer-by-Layer

The Layer-by-Layer method, also called the Langmuir–Blodgett deposition method, occupies an important space within the range of possibilities for producing multilayer films. This fact is related especially to its low cost, simple operation, and versatility [32]. This methodology consists of electrostatic interaction between polymers with opposite charges. This technology possibly produces layers through the immersion of the substrate in a polycation solution and after the immersion in a polyanion solution [33,34]. The main advantages of this technology are related to the production of uniform multilayer films and the possibility of controlling the thickness and obtaining it at the nanoscale [33,34,35]. The Layer-by-Layer deposition mechanism can be observed through the schematic illustration proposed by De Villiers et al. [36], who divided this mechanism into four steps: step A corresponds to the immersion of the substrate in a polycation, step B represents washing in solvent for polycation, step C represents the immersion of the substrate in a polyanion, and finally step D represents washing in solvent for polyanion. These steps can be repeated n times to obtain the multilayer system, represented by the letter E. This schematic process can be visualized in Figure 1.

The literature on this technique is extensive, and some works can be described below. Fabra et al. [37] developed nano-laminated films using sodium alginate and zein nanocapsules containing carvacrol to improve the shelf life of packaged foods. Koca and Bayramoglu [38] also used this technique to develop a material that can be used as food packaging, but using lysozyme with iota-carrageenan and gum arabic, and found an improvement in the barrier properties of the product. Wang and Zhang [39] also used the protein lysozyme but alternated it with sodium alginate and electrospun cellulose acetate and evaluated the antibacterial activity of this material in milk.

Although promising, the Layer-by-Layer technique has not yet found acceptance in the industry to produce multilayer films. This fact is mainly due to the difficulty of reliable upscaling and material waste during the coating process. The last challenge becomes even more important when it uses valuable materials. Thus, the Layer-by-Layer method presents encouraging results for producing multi-layered food packaging materials capable of prolonging the shelf life of packaged foods. However, this method still lacks a detailed study of its industrial viability [37,40,41].

### 3.2. Casting

The casting method is the most common and one of the oldest monolayer thin film-forming operations used on the laboratory scale [42,43]. It is a colloidal solution composed of a polymer, solvent, and usually a plasticizer. First, the colloidal solution must be poured onto the appropriate support and then dried. Drying conditions depend mainly on the relationship with the physicochemical properties of the polymer [32,44]. During the dissolution of the colloidal solution, the solvent is evaporated. With this, a polymer concentration occurs, which causes the aggregation of the molecules, enabling a three-dimensional structure [45]. For multilayer film production by the casting method, the polymer solution should be poured into another polymeric layer previously prepared and dried [32,46]. Figure 2 shows the casting method’s schematic process of multilayer film formation.

The main advantage of the casting method is the simplicity of the process. However, as it contains a large amount of liquid in the film-forming solution, the drying step is the most time consuming, demanding a large energy consumption [47,48]. Besides, other factors can be cited as obstacles in the casting technique, such as bad format options (simple plates and tubes are usually produced) and difficulty scaling up. The latest is especially due to the impact on film characteristics caused by differences in variables such as temperature and air speed [49].

Despite this, the literature on multilayer film formation using the casting technique is extensive. For example, Haghighi et al. [7] elaborated a bilayer film based on chitosan and gelatin using the casting method. On the other hand, Cerqueira et al. [50] developed a biodegradable multilayer system based on alginate and zein polymers joining the casting and electrospinning methodologies. Along the same line, Figueroa-Lopez et al. [22] also used electrospinning and casting methods to produce a multilayer system based on gelatin and polycaprolactone.

### 3.3. Compression Molding

Another way to produce thin films is the compression technique. This method consists of heating and then cooling. First, the pressure is applied in the fusion phase until the cool-down phase [48,51]. This process occurs in a compression molding press composed of two heating plates, the bottom plate where the polymer is added [32,51]. The compression method has the advantages of simplicity and less time consumption that allow production on a large scale. However, compression molding processes normally use temperatures around 180 °C, making the use of natural additives unfeasible (since these are thermosensitive or even volatile) in the polymer matrix [52,53,54]. Despite that, according to Siqueira et al. [48], the use of this method for the production of bilayer films is increasing, mainly due to the possibility of aggregating properties of different polymers in a single material. Figure 3 shows a schematic of the compression method’s bilayer film formation process.

The material obtained by this process has the potential for use in several sectors, including the packaging sector, where it can incorporate additives to confer active properties to the final product [32,55]. Fabra et al. [56] reported the development of bilayer films based on wheat gluten and whey protein isolate, wheat gluten and soy protein isolate-guar gum, wheat gluten, and zein. For all bilayer films, the authors combined the techniques of compression molding (to prepare the wheat gluten layer) and electrospinning (to prepare the second layer with the remaining polymers), producing bilayer films with and without active compounds. Andrade et al. [21] described the development of multilayer films based on PLA and PVA polymers with and without additives (carvacrol, lecithin encapsulated carvacrol, or ferulic acid) using the compression molding technique. Ordoñez et al. [26] also developed multilayer films using the compression molding technique. The authors used starch and PLA as biopolymers and incorporated cinnamic and ferulic acids to confer antibacterial properties on the final product.

For Andrade et al. [21] and Ordoñez et al. [26], the way to overcome the thermal degradation of bioactive compounds was to combine the casting method with compression molding since the casting methodology normally uses lower temperatures. Thus, the potential of using the technique to produce active multilayer films is evident.

### 3.4. Extrusion

Extrusion is also a food packaging manufacturing process that enables the processing of different materials. This mechanism forces the material to pass through the matrix, at a specific rate, under different mechanical and temperature conditions. A high-pressure drop in the matrix causes water evaporation and consequently promotes nucleation. Finally, this process causes a significant product expansion in the extruder exit [57,58]. This process can produce multilayer films using two or more different polymers extruded together. Figure 4 shows a schematic of the extrusion method’s process of bilayer film formation.

Some examples can be found in the literature, such as the work by Pant et al. [57] that reported using extrusion combined with lamination and thermoforming. The food packaging material was based on bio-based linear low-density polyethylene and PLA and incorporated with gallic acid. Granda-Restrepo et al. [58] have also developed multilayer films using the extrusion technique. According to the authors, the multilayer films were obtained with different polymers (high-density polyethylene, ethylene vinyl alcohol, and low-density polyethylene) and incorporated with different antioxidant agents (butylated hydroxytoluene, butylated hydroxyanisole, and alpha-tocopherol) for promising use as food packaging.

### 3.5. Electrospinning

Electrospinning is an innovative and promising technique that presents the possibility of producing fibrous materials in submicron or nanoscale as the main advantage. This fact results in a final material with a high surface-to-mass ratio, porosity, and encapsulation efficiencies of bioactive compounds Patil et al. [59]. This technology makes use of electrostatic forces to form fibrous materials. Furthermore, the mechanism consists of extruding a polymeric solution from the spinneret, forming a drop at the tip of the needle that is connected to a high-voltage power supply. This drop undergoes a deformation when the electric field is applied until the surface tension is overcome. When this happens, the polymer jet is ejected and deposited in the collector, which must be grounded [60].

Another important fact in the electrospinning technique is the parameters involved in the process that influence the final product. These parameters include apparent viscosity, surface tension, and electrical conductivity [61]. Figure 5 shows a schematic of the electrospinning process.

The electrospinning technique shows some advantages, such as low cost, high surface area, and capacity to produce continuous fibers. On the other hand, the technique’s main disadvantage is the jet’s instability. However, this problem can be overcome by adjusting the electrospinning parameters (mentioned above) [59]. Therefore, several authors have reported studies using the electrospinning technique.

Arkoun et al. [60] reported the study of the efficiency of maintaining quality safety of red meat packed with a multilayer system composed of chitosan/poly(ethylene oxide) electrospun fibers deposited in conventional food packaging. As a result, the authors proved that the multilayer system developed could extend the product’s shelf life by one week. Wang et al. [62] used the electrospinning technique to produce a multilayer system based on ethylcellulose nanofibers as the outer layer and gelatin nanofibers embedded with curcumin as the inner layer. As a result, the authors achieved a controlled release of the active compound during 96 h, superior to the release of the same compound contained in the gelatin monolayer film that occurred for 30 min. As previously mentioned, the electrospinning technique can be combined with other methods to develop a multilayer system. In this sense, Estevez-Areco et al. [11] developed an active bilayer film based on thermoplastic starch and ZnO nanorods as the outer layer and poly(vinyl alcohol) fiber incorporated with rosemary polyphenols as the inner layer. Thus, electrospinning emerges as an interesting alternative for producing multilayer systems that can be applied to food packaging.

## 4. Main Natural Sources of Bioactive Compounds

Different sources of bioactive compounds are present in nature, such as vegetables, fruits, cereals, and their by-products. These natural resources are characterized by health benefits and antimicrobial and antioxidant properties due to their chemical composition and can be applied in nutraceuticals, cosmetics, and food packaging sectors [63,64,65]. The compounds mentioned are primarily the result of secondary metabolites and, in some cases, primary metabolites and can be classified as phenolic acids and polyphenols, carotenoids, alkaloids, terpenes and terpenoids, tannins, anthocyanins, flavonoids, fatty acids, and lipids, for example [66,67]. Thus, the search for natural resources, with benefits like the ones mentioned above, is on the rise and has wide applicability [68].

In the pharmaceutical industry, natural compounds play an important role, as they can be an alternative to synthetic drugs, thus reducing the side effects caused by them [69]. The reason for using natural sources in the cosmetic industry is similar: the reduction of side effects and low toxicity of compounds combined with greater acceptance of products by consumers [70]. In line with this, for food packaging, natural bioactive compounds are linked to their low toxicity and ability to provide functional properties to materials [71]. For this reason, there is increasing interest in reducing synthetic additives, which natural additives can substitute. Many works have demonstrated the potential of using natural additives in food packaging materials with different properties. Table 2 shows recent works reported in the literature on natural additives in food packaging and their different functionalities.

According to Table 2, it is possible to identify a high tendency among researchers to use industrial and agro-industrial by-products as a source of bioactive compounds. With this, the circular economy is favored, as these materials can receive an appropriate destination and possibly add value to waste. Therefore, the circular economy can be defined based on the goals of reduction, reuse, recycling, redesign, remanufacturing, and recovery. This concept is extremely relevant since natural resources are limited. Furthermore, the accumulation of solid waste is a dangerous environmental problem [82,83,84].

Although they are abundant sources, using natural bioactive compounds in packaging systems is still a challenge when seeking standardization of the final product, mainly due to variation in the composition of natural sources. This fact is directly related to the conditions adopted in agricultural practices and the maturity degree of the plant. Moreover, once extracted and purified, the bioactive compound becomes more susceptible to degradation and volatilization, especially when, in liquid form, exposed to light, pH, oxygen, and others. Therefore, some procedures are recommended to preserve phytochemicals, such as drying and encapsulating these compounds. When encapsulated, bioactive compounds can improve their stability through a release system composed of a coating agent. On the other hand, drying is also a good option to preserve phytochemicals. Among the various possible drying methods, spray-drying and freeze-drying perform an important space in the food industry [79,85].

Despite being treated previously, using bioactive compounds as additives in food packaging still faces some difficulties. These are especially due to the high sensitivity of natural compounds against temperature, oxygen, and uncontrolled release [21,86]. In this sense, multilayer packaging systems can encapsulate natural bioactive compounds, reducing their losses and providing active properties for this product.

## 5. Multilayer Systems as Active Biodegradable Food Packaging

The interest in developing natural materials for food packaging with particular properties, such as antimicrobials, antioxidants, and antifungals, brought a challenge in using natural compounds that add such properties. As an emerging technology, the multilayer packaging system appears as a viable strategy to offer active properties to the material derived from natural raw materials, reducing the losses of these compounds [87]. Another important point in this type of packaging is the controlled release of bioactive compounds that are allowed through the structure design of layers. According to Nogueira et al. [88] and Chen et al. [87], this structure is composed of:An external layer with a high barrier property. This layer prevents or reduces the bioactive compounds lost from the polymeric matrix to the environment;An intermediate layer with a high diffusion property. This layer contains the bioactive compound;An internal layer (in contact with food) with less swelling index than the intermediate layer. This layer has the function of controlling the release rate of active agents for packaged food.

Figure 6 shows, in schematic form, the structures of a multilayer package. With this configuration of the biodegradable packaging system, it is possible to obtain a positive interaction between packaging and food without the active agent is in direct contact with the food [89]. It is important to highlight the possibility of inserting nanotechnology, as an example of nanofibers, in designing a multilayer system. The best layer in which to use nanofiber is not yet unanimously agreed upon among researchers and is frequently applied in the external layers [22] or the middle layer [50]. Thus, by adding a nanofiber to the multilayer design, it is possible to obtain a final product with improved mechanical and barrier properties and the excellent encapsulation capacity of the bioactive compounds [90,91]. According to Monção et al. [92], in the last 10 years, new active food packaging development has increased, with a considerable number referring to multilayer systems.

Generally, this type of multilayer packaging is composed of different polymers, making it possible to add multiple functionalities to the product [15]. However, the choice of biopolymers that should be included in the packaging system depends on what is expected of the final product and the characteristics of the food to be packaged [93].

Some works can be cited as examples of multilayer systems applied to food packaging, such as the study developed by Ordoñez et al. [26]. They created a multilayer film based on starch and PLA polymers incorporated with cinnamic and ferulic acids. The authors justified using the multilayer film due to the susceptibility to moisture of the starch film. However, this characteristic is not interesting when material for application in food packaging is desired, mainly due to its degradability. One way to solve this problem is by adding an extra layer with good barrier properties, as in the case of PLA polymer.

A similar case was reported by Cai et al. [23], who developed a PLA/gelatin-curcumin/PLA-based multilayer packaging system. Gelatin is a good film former and has important properties such as presenting a good barrier to light and oxygen. However, as it has a hydrophilic profile, the authors opted to create a system composed of a gelatin interlayer containing the bioactive compound and two outer polycaprolactone layers with good mechanical properties and a hydrophobic profile.

Chen et al. [87] developed a multilayer system based on polypropylene and poly (vinyl alcohol) to control the release of polyphenols since the release rate of bioactive compounds in the poly (vinyl alcohol) monolayer system was very fast, especially due to their high hydrophilicity. For this reason, the authors described using the internal polypropylene microporous film, the intermediate poly (vinyl alcohol) layer containing the bioactive compound, and the external polypropylene layer without pores.

Furthermore, research about using multilayer systems for active food packaging is increasing. Using a structure composed of more than one layer allows the development of active packaging material with controlled release of the active compounds. Consequently, it is possible to improve the shelf life of food products since the active agents show positive interaction with packaged food, reducing their deterioration during storage [20].

Xia et al. [20] have developed a multilayer system consisting of three layers (zein, zein/gelatin, zein) and added tea polyphenols in the middle and inner layers. When exposed to fresh fruits, these films showed antioxidant properties, delaying their browning. Furthermore, the authors also reported the controlled release of the antioxidant property of the multilayer films once half of the monolayer films shortened the release ratio.

Therefore, the literature on multilayer food packaging is vast, and the choice of polymers that will compose the packaging system requires an improved study involving the desired functions for the final product as well as the characteristics of the polymers.

## 6. Positive and Negative Aspects of Multilayer Packaging Development

Multilayer packaging systems consist of a combination of different materials that will make up the layers. This configuration can bring many benefits to the packaging system and thereby improve the quality and safety of the packaged material. In addition, combining different materials in a single structure makes it possible to add the individual properties of polymers to the final product. This fact is due to the difficulty of obtaining all requirements of food packaging in a monolayer packaging structure. Added to that, when it comes to biopolymers for packaging materials, this difficulty intensifies. Unfortunately, monolayer packaging films formed from biopolymers are not competitive with synthetic structures, requiring new alternatives that make them economically attractive. Therefore, the development of multilayer systems is an alternative capable of improving properties such as high barrier and mechanical resistance, for example.

On the other hand, the literature reports some disadvantages related to using this methodology, such as a long production period, since it occurs in two or more stages, resulting in high energy consumption [47,48,94]. In addition, the multilayer method presents an important challenge related to adhesion between the layers of the system. This problem occurs when the chosen biopolymers have different profiles (hydrophilic/hydrophobic) [50,95]. Thus, the importance of studying the best operating conditions and the characteristics of the polymers is evident.

## 7. Trends and Challenges in the Use of Multilayer Systems

The food packaging industry is constantly evolving, and in recent years it has paid special attention to biodegradable films. This fact is related to the negative environmental impact caused by the indiscriminate use of synthetic polymers. It is estimated that food packaging is responsible for approximately 40% of the total plastic produced, and each food packaging material requires a very high decomposition time, around one hundred years [21,96,97,98]. For this reason, efforts to develop biopolymeric food packaging are highly essential.

On the other hand, food packaging produced with biopolymers still has some disadvantages compared to synthetic polymer packaging, making their commercialization difficult. Generally, this type of packaging has low mechanical and barrier properties. In this context, developing multilayer packaging for food is an effective strategy for packaging using biopolymers to be economically competitive with synthetic ones [32,97].

The development of active food packaging is linked to the development of multilayer packaging systems. Currently, approaches that use residues as agro-industrial by-products have received much attention both for the development of biopolymers and for obtaining natural additives. Therefore, this type of packaging is a trend in the food packaging industry. Thus, with the concepts of multilayer packaging and active food packaging, it is possible to achieve a controlled release system of active compounds that is not achievable in a monolayer system [21,82].

Another point is that nanotechnology is promising to contribute to controlling the release of active compounds in multilayer packaging. In addition, nanostructures improve other important properties such as presentation of mechanical, thermal, water, and oxygen barriers [99]. In recent years the electrospinning technique has received much attention due to the possibility of developing a material that can be deposited on the surface of a film, forming a bilayer or multilayer system with unique properties [100].

Despite the eco-friendly appeal of biopolymeric packaging, its commercialization is still a limiting factor, with few options on the market. Most biodegradable food packaging is still in the research and development stage. For the economic viability of this product, some challenges must be overcome, such as the price of equipment required to produce packaging. In this case, a detailed study of the foods that will be packaged is of paramount importance to ensure the cost-effectiveness of the final product [101,102]. In line with this, increasing the scale of production of biopolymer materials is a problematic and limiting point for their commercialization. This is especially due to most research in the area reporting the use of the casting method for manufacturing. The obstacle to be overcome when using this methodology is related to the amount of water that must be evaporated and the energy demand necessary for this [48,103].

Thus, while there are challenges to be faced and advances to be made in achieving substantial commercialization of food packaging based on biopolymer materials, the use of the same at the expense of petrochemicals is extremely important for the maintenance of the environment. In this sense, the present work is an important tool for developing new materials with a sustainable bias.

## 8. Conclusions

The constant change and the demand for better food packaging materials with environmental and human health concerns have gained great research attention, evidenced by the increasing number of publications in this area. In addition, Multilayer systems as an alternative to improve the performance of the developed materials are also strongly discussed. However, research does not address in depth the biopolymers that can be used in developing these materials, nor the unique properties that can be conferred on the final material by adding natural additives.

In this sense, it is worth emphasizing the importance of studies on agro-industrial wastes as a source of bioactive compounds. Thus, it is important to know in advance the characteristics of the natural additive and the biopolymer and choose the best method of obtaining the multilayer system to meet the required properties of a food packaging material. In addition, as important as the method of obtaining the multilayer system is the choice of the combination of biopolymers that will constitute the layers of this system.

Thus, advances in the food packaging industry are taking place toward using raw materials with low or no negative impact on the environment and human health, in addition to increasingly technological materials, with an emphasis on active packaging. This statement is valid for polymeric and additive sources. However, from another point of view, the economy continues to be important. Therefore, alternatives that enable the use of residues, such as agro-industrial waste with low or no commercial value, are recommended.

## Figures and Tables

**Figure 1 foods-12-01692-f001:**
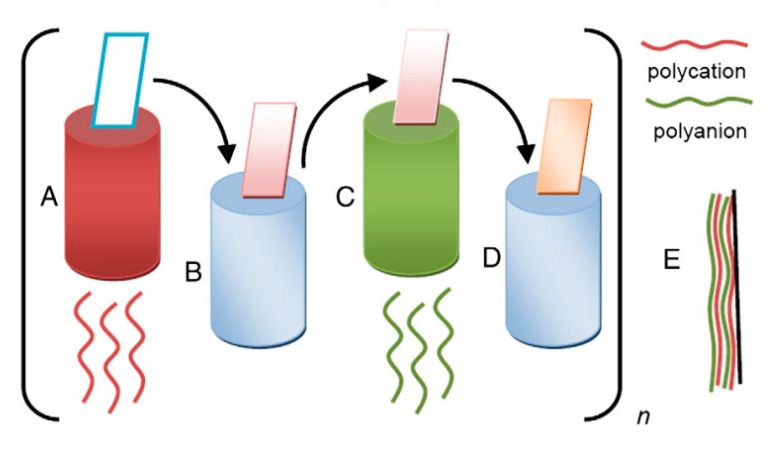
Layer-by-Layer schematic process [36].

**Figure 2 foods-12-01692-f002:**
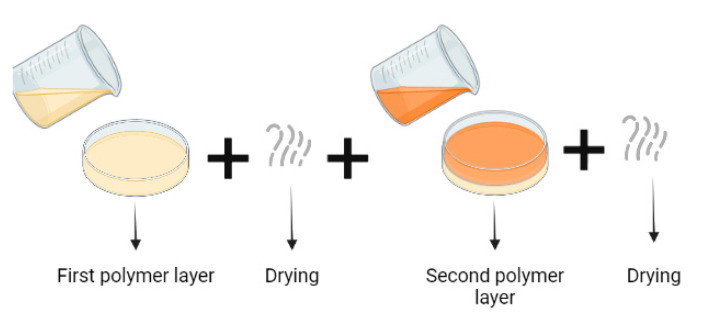
Schematic process of multilayer film formation by the casting method.

**Figure 3 foods-12-01692-f003:**
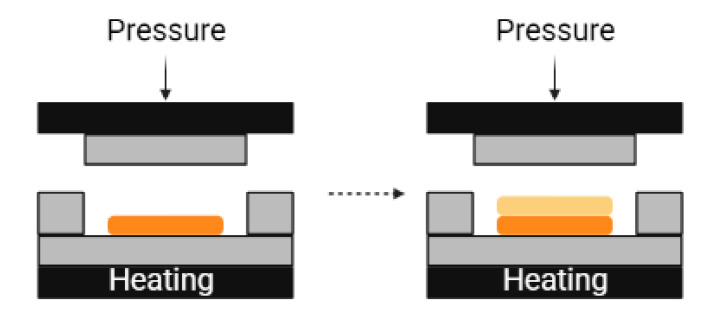
Schematic of process of bilayer film formation by the compression method.

**Figure 4 foods-12-01692-f004:**
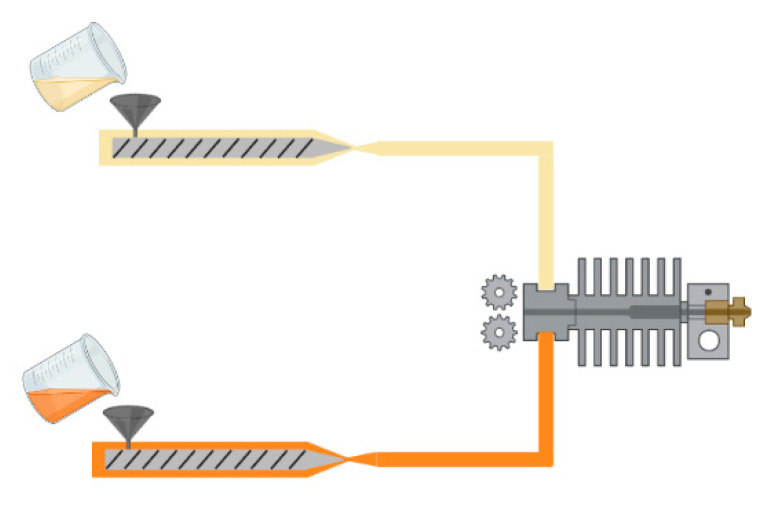
Schematic of process of bilayer film formation by the extrusion method.

**Figure 5 foods-12-01692-f005:**
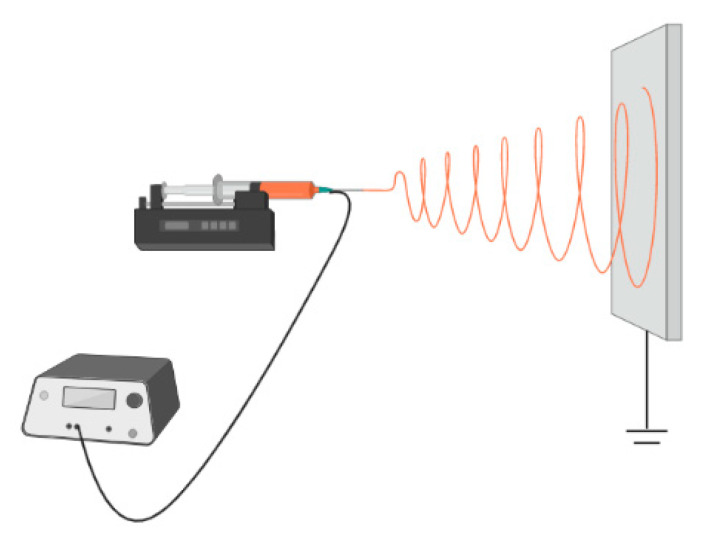
Schematic of the electrospinning process.

**Figure 6 foods-12-01692-f006:**
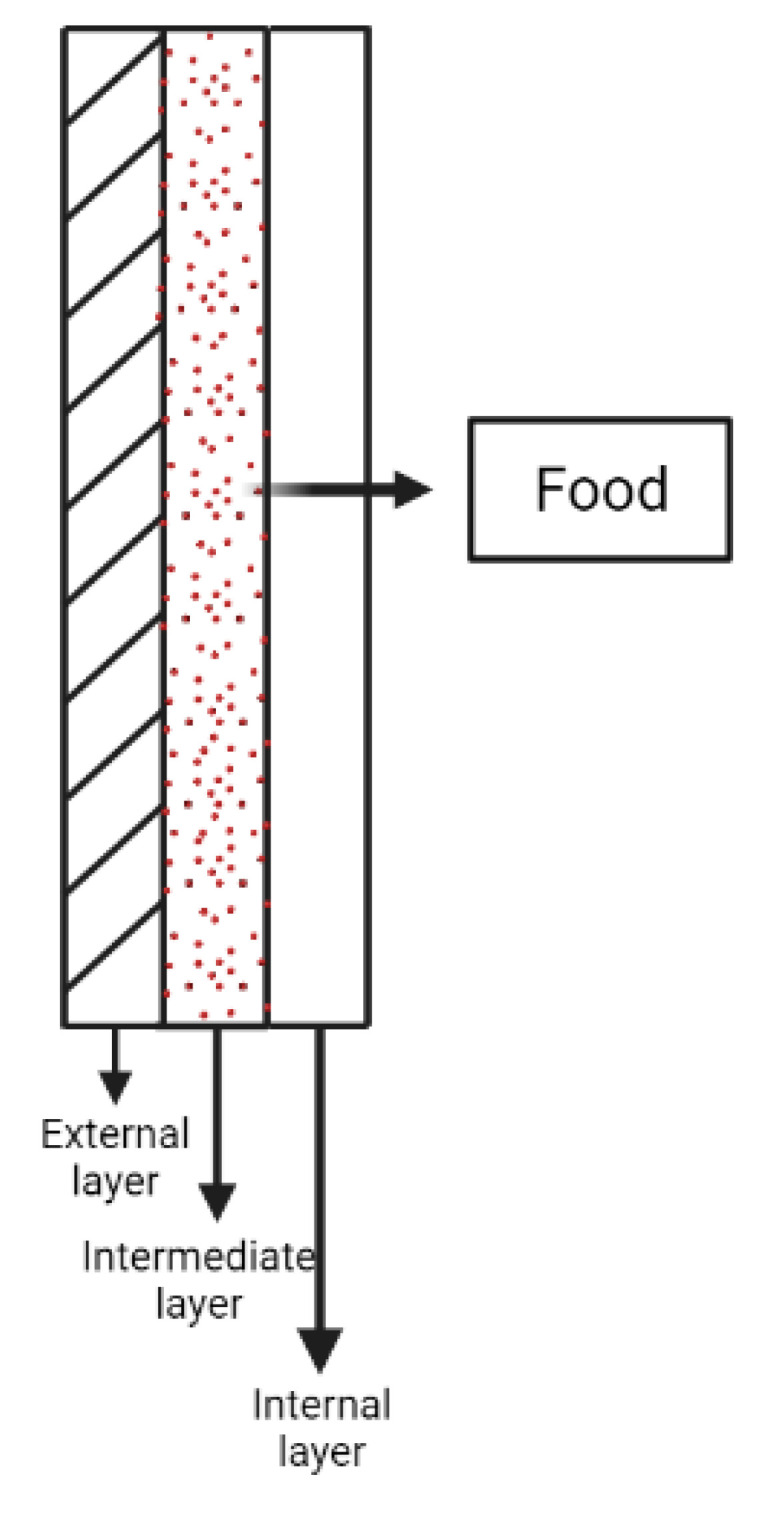
Multilayer packaging structure.

**Table 1 foods-12-01692-t001:** Functionalities of multilayer food packaging.

Source of Active Compound	Polymer	Packaging Function	Reference
Black Pepper Oleoresin	Polycaprolactone Gelatin	Antimicrobial multilayer food packaging	Figueroa-Lopez et al. [22]
ZnO nanoparticles and carvacrol	Polylactic Acid Chitosan (2-Carboxyethyl)-β-Cyclodextrin	Antimicrobial multilayer food packaging	Andrade-Del Olmo et al. [17]
Carvacrol and ferulic acid	Poly(Vinyl Alcohol) Polylactic acid	Antimicrobial multilayer food packaging	Andrade et al. [21]
Curcumin	Polycaprolactone Gelatin	Antimicrobial multilayer food packaging	Cai et al. [23]
Clove essential oil	Gelatin Myofibrillar protein	Antioxidant multilayer food packaging	Jiang et al. [24]
Tea polyphenol	Zein Gelatin	Antioxidant multilayer food packaging	Xia et al. [20]
Gallic acid	Polylactide	Antioxidant multilayer food packaging	Quiles-Carrillo et al. [25]
Cinnamic and Ferulic acids	Starch Polylactic acid	Antimicrobial multilayer food packaging	Ordoñez et al. [26]

**Table 2 foods-12-01692-t002:** Recent works reported in the literature on natural bioactive sources.

Bioactive Compound Source	Natural Additive Function	Reference
Açaí (*Euterpe oleracea*)	Colorimetric pH sensors	Silva et al. [72]
Mulberry	Antioxidant and colorimetric pH sensors	Liu et al. [73]
Grape skin	Colorimetric pH sensors	Chi et al. [74]
Bay and Sage leaves	Antioxidant activity	Oudjedi et al. [75]
Prunus maackii	Antioxidant and colorimetric pH sensors	Sun et al. [76]
Olive leaves	Antioxidant activity	da Rosa et al. [77]
Jaboticaba peel	Antioxidant and antimicrobial activities	Avila et al. [78]
Coconut shell	Antioxidant activity	Tanwar et al. [79]
Pinhão coat extract (*Araucaria angustifolia*)	Antioxidant activity	Fonseca et al. [80]
Mango leaf	Antioxidant and antimicrobial activities	Bastante et al. [81]
Artemisia campestris	Antioxidant activity	Moalla et al. [71]

## Data Availability

Data is contained within the article.

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
