# Peer review of "Trends in Bioactive Multilayer Films: Perspectives in the Use of Polysaccharides, Proteins, and Carbohydrates with Natural Additives for Application in Food Packaging"

_foods, 2023, doi:10.3390/foods12081692_

Round 1

Reviewer 1 Report

There have been many reports on active packaging materials in general review, so this paper is rarely novelty. An in-depth review of a particular packaging substrate, such as polysaccharides, proteins and carbohydrates or preparation method is more fascinating.

Author Response

Reviewer #1

There have been many reports on active packaging materials in general review, so this paper is rarely novelty. An in-depth review of a particular packaging substrate, such as polysaccharides, proteins and carbohydrates or preparation method is more fascinating.

RESPONSE: Thanks to the reviewer for appreciating our effect. We agree with the number of reports on active packaging in the literature. However, this number is considerably reduced when it comes to a multilayer active biopolymer food packaging system. Therefore, the innovative character of this work is related to a deep and concise approach not only to the methods of obtaining a multilayer food packaging system but also to the possibility of using biopolymers and natural additives, with an emphasis on wastes and industrial by-products. To better expose this purpose, we added the topic “conclusions” where we address the main points of the manuscript highlighting the trends and needs of the food packaging industry and thus highlighting the innovation of this manuscript.

Reviewer 2 Report

Dear authors,

this article gives an overview on the perspectives in the use of biopolymers with bioactive additives in multilayer films for the application in food packaging. The paper gives us an update on the current research of multilayer films based on biopolymers, in the production of the packaging material, the main natural sources of bioactive compounds and strategies for producing biopolymer. In addition, the applications of these materials and trends in this sector were highlighted.

On the one hand the central theme could be more focused if you omit to focus on “biodegradable” in the abstract and the introduction, since this topic is of only subsidiary interest in the following chapters. On the other hand your review clearly focuses on bioactive additives. Therefore the adjective “bioactive” should be mentioned in the title of your paper.

Kind regards

Author Response

Reviewer #2

This article gives an overview on the perspectives in the use of biopolymers with bioactive additives in multilayer films for the application in food packaging. The paper gives us an update on the current research of multilayer films based on biopolymers, in the production of the packaging material, the main natural sources of bioactive compounds and strategies for producing biopolymer. In addition, the applications of these materials and trends in this sector were highlighted.

On the one hand the central theme could be more focused if you omit to focus on “biodegradable” in the abstract and the introduction, since this topic is of only subsidiary interest in the following chapters. On the other hand your review clearly focuses on bioactive additives. Therefore the adjective “bioactive” should be mentioned in the title of your paper.

Kind regards

RESPONSE: We thank you the reviewer for appreciating our effect. We modified the manuscript and title as suggested.

Reviewer 3 Report

//

The duty of review manuscript  are the authors must be providing detail information to the reader other than a large overview of the topic they are reporting on and, also a guide for the reader towards a future where the field is headed.

suggest to get english proofreader

Author should highlight the best conclusions

Author Response

Reviewer #3

The duty of review manuscript  are the authors must be providing detail information to the reader other than a large overview of the topic they are reporting on and, also a guide for the reader towards a future where the field is headed.

suggest to get english proofreader

Author should highlight the best conclusions

RESPONSE: Thanks to the reviewer for appreciating our effect. Throughout the manuscript, we sought to provide an overview of the theme and future perspectives, especially in item “7. Trends and challenges in the use of multilayer systems”. However, to clearly and better provide a guide for the reader toward a future of multilayer films, we have added a new topic (conclusions) where we address the main points of the manuscript with emphasis on the trends and needs of the food packaging industry, as suggested. The English was also improved.

Reviewer 4 Report

Paper: Trends in multilayer films: perspectives in the use of polysaccharides, proteins and carbohydrates with natural additives for 3 application in food packaging

 Manuscript ID: foods-2185833.

 Recommendation: minor revisions

 Comments: The work is interesting and well structured. It provides a wide analysis of multilayer films for packaging applications. The authors should just add a short paragraph on the main physical properties of these natural materials such as transparency, thermal properties, solubility, permeability…

Author Response

Reviewer #4

Manuscript ID: foods-2185833.

Recommendation: minor revisions

Comments: The work is interesting and well structured. It provides a wide analysis of multilayer films for packaging applications. The authors should just add a short paragraph on the main physical properties of these natural materials such as transparency, thermal properties, solubility, permeability…

RESPONSE: We thank you the reviewer for appreciating our effect. We modified the manuscript by adding a paragraph on the main properties of biopolymer materials for food packaging, as suggested.

Round 2

Reviewer 3 Report

no comments